# Predicting Dyslexia in Adolescents from Eye Movements during Free Painting Viewing

**DOI:** 10.3390/brainsci12081031

**Published:** 2022-08-03

**Authors:** Alae Eddine El Hmimdi, Lindsey M Ward, Themis Palpanas, Vivien Sainte Fare Garnot, Zoï Kapoula

**Affiliations:** 1Orasis Eye Analytics and Rehabilitation, CNRS Spinoff up, 12 rue Lacretelle, 75015 Paris, France; alae-eddine.el-hmimdi@etu.u-paris.fr (A.E.E.H.); vivien.sainte-fare-garnot@polytechnique.edu (V.S.F.G.); 2LIPADE, French University Institute (IUF), Laboratoire d’Informatique Paris Descartes, University of Paris, 45 Rue Des Saints-Pères, 75006 Paris, France; themis@mi.parisdescartes.fr; 3IRIS Lab, Neurophysiology of Binocular Motor Control and Vision, CNRS UAR 2022, University of Paris, 45 rue des Saints Pères, 75006 Paris, France; lward@mednet.ucla.edu

**Keywords:** saccades, vergence, eye movement, paintings free exploration, machine learning, dyslexia, feature engineering

## Abstract

It is known that dyslexics present eye movement abnormalities. Previously, we have shown that eye movement abnormalities during reading or during saccade and vergence testing can predict dyslexia successfully. The current study further examines this issue focusing on eye movements during free exploration of paintings; the dataset was provided by a study in our laboratory carried by Ward and Kapoula. Machine learning (ML) classifiers were applied to eye movement features extracted by the software AIDEAL: a velocity threshold analysis reporting amplitude speed and disconjugacy of horizontal saccades. In addition, a new feature was introduced that concerns only the very short periods during which the eyes were moving, one to the left the other to the right; such periods occurred mostly during fixations between saccades; we calculated a global index of the frequency of such disconjugacy segments, of their duration and their amplitude. Such continuous evaluation of disconjugacy throughout the time series of eye movements differs from the disconjugacy feature that describes inequality of the saccade amplitude between the two eyes. The results show that both AIDEAL features, and the Disconjugacy Global Index (DGI) enable successful categorization of dyslexics from non-dyslexics, at least when applying this analysis to the specific paintings used in the present study. We suggest that this high power of predictability arises from both the content of the paintings selected and the physiologic relevance of eye movement features extracted by the AIDEAL and the DGI.

## 1. Introduction

Eye movement abnormalities are known to exist in dyslexics. Recently, we have shown that eye movement abnormalities are intrinsic and not just a consequence of reading difficulty [1]; in another study, Ward and Kapoula reported eye movement abnormalities during reading of two different types of text: a senseless text requiring decoding word by word, and a normal text with meaning [2]. The authors reported increased eye movement abnormalities in the former case and suggested that reading text difficulty can exacerbate an otherwise fragile eye movement control system. Namely, the physiologic features that were found to be systematically poor in dyslexic teenagers were the slow average velocity of eye movements despite the normal initial velocity of the movements, longer duration, and increased discoordination between the two eyes as measured by the disconjugacy feature. Therefore, there is an interaction between the type of reading text and the eye movement abnormalities [2].

Asvestopoulou et al. [3] reported evidence for the predictive capacity of eye movement abnormalities during reading texts and the dependency of the predictive power on the type of reading text. In a more recent study, El Hmimdi et al. [4] investigated predicting dyslexia using a set of linear and nonlinear classifiers, by considering all eye movement features (latency, peak and average velocity, duration, amplitude, disconjugacy during post-saccadic drifts or of the saccade itself, etc.); those features were extracted automatically by the software AIDEAL from different sets of data produced by Ward and Kapoula [1,2], namely, those of saccades and vergence eye movements to single LED targets presented by the REMOBI device (patent WO2011073288), and the two-reading texts mentioned above. This study showed for the first time that eye movement abnormalities per se in the REMOBI tests, independently from reading, can successfully predict dyslexia and even the reading speed, with an accuracy greater than 81%, on a balanced dataset. The most predictive features were particularly, the velocity, the duration, and the disconjugacy. Moreover, the predictive capacity for categorizing dyslexics and non-dyslexics was better for the eye movement abnormalities from the meaningless reading text rather than from the meaningful text.

During reading, eye movements are self-generated and are a kind of free exploration of the text where the tested person adjusts the rhythm of the sequence of saccades and fixations at will. While in the REMOBI testing, eye movements are the immediate responses to external stimulation (space and time) programmed with this device.

The present study uses the data from another study [5], in which dyslexic and non-dyslexic teenagers spontaneously explore a selection of paintings. The study focused on free exploration eye movements, but outside of text and reading contexts. Would eye movement abnormalities show up even in ludic conditions? The authors have chosen specific types of paintings, amongst others Op-Art from Bridget Riley, creating the illusion of visual movement in-depth.

Also, paintings from Escher and Magritt, because of their multiple impossible spaces or hidden spaces, or because of contradictions between image and text [6]. According to these authors, pictorial movement perception and pictorial space perception would be connected with theories and controversies concerning increased creativity in dyslexia.

The main hypothesis is that some of the eye movement deficits known to exist in dyslexia, e.g., disconjugate saccades and disconjugate drifts during fixations could, in the context of painting viewing, be beneficial. Particularly, it has been suggested that the illusion of movement from op-Art is reinforced by micro-movements and fixational instability such as accommodation changes [7].

In such cases, one would expect increased subjective movement illusion in dyslexics. Concerning the paintings from Magritt or Escher with multiple hidden or bistable spaces, or with contradictions between image and words,) Ward and Kapoula, aimed to substantiate the hypothesis debated by some according to which dyslexics have a better capacity in extracting form from the background and perceiving space [8].

Using ML, the purpose of the present study is to go further and examine the predictive value of eye movement characteristics during free viewing of these two groups of paintings challenging the perception of visual motion or perception of pictorial spaces.

As in the El Hmimdi et al. [4] paper, the goal of the present study is to bring the quantitative power of the ML models for predicting dyslexia, to gain new insight from the data. In this study, we’ve trained linear and nonlinear models on the data sets coming from carefully designed and hypothesis-driven neuro-scientific studies.

Moreover, the present study introduces an additional feature, DGI that detects all segments of the eye movement traces during which the two eyes are moving in opposite directions. Such segments might be very brief in duration and can occur throughout the signal, i.e., during the saccades or the post-saccadic fixations or the resting period between large movements.

The motivation behind developing such scores is to capture the reaction of the different subjects to the 3D illusion optics present in the seven paintings in terms of motion or space composition complexity.

The results demonstrate that when considering the seven painting dataset separately, the predictive power of the features derived from AIDEAL depends on the type of painting, while the DGI is more universal and predictive no matter what the painting is.

However, as soon as the dataset size grew, the ability of AIDEAL, based on deterministic eye movement features, to predict dyslexia became more competitive. Further, the complementarity between the two sets of features is investigated.

## 2. Materials and Methods

### 2.1. Participants

The data used for the ML analysis is taken from published studies, please see Ward and Kapoula for full details regarding methods and materials [1,2,5]. In brief, eye movements were recorded during painting exploration in 46 dyslexic adolescents (18 females, 28 male; mean age 15.52, SD 2.45) and 41 non-dyslexic adolescents (20 female, 21 male; mean age 14.78 +/− 2.44) recruited from schools in Paris. Dyslexic adolescents were diagnosed in specialized medical centers and were admitted to their schools based on their dyslexia diagnosis.

Typically, diagnosis by these centers includes multiple testing which confers a diagnosis of visual, phonological dyslexia, mixed dyslexia, etc. We, therefore, did not have children self-identify their condition. Both dyslexic and non-dyslexic adolescents had no known neurologic or psychiatric abnormalities. Typical readers had no history of reading difficulty, no visual impairment, or difficulty with near vision. Of dyslexics, 34.0% (16/47) identified that their primary issue was visual/reading based, 4.3% (2/47) auditory, 2.1% (1/47) writing, and 59.6 (28/47) were mixed or unknown.

The reading speed was measured with eye movement recording by Ward and Kapoula [2], using the Alouette test; reading speed was significantly lower and more variable for dyslexics than non-dyslexics. (median ± std was 92 ± 109 for the dyslexic and 136 ± 29 for the non-dyslexic population). The full text of the Alouette test is presented in Figure A1.

The investigation adhered to the principles of the Declaration of Helsinki and was approved by our Institutional Human Experimentation Committee (CPP CNRS 18 011). Written consent was obtained from the adolescents and/or their parents after the experimental procedure was explained. The tests were conducted by two research assistants, who were trained together using the same material and conducted the experiment together for each measurement.

### 2.2. Eye Movement Recording Device

For each adolescent, eye movements were recorded binocularly with a head-mounted video-oculography device, Pupil Core [9], enabling binocular recording at 200 Hz per eye (Pupil Labs, Berlin, Germany).

Subsequently, they were analyzed with AIDEAL, (patent PCT/EP2021/062224). For saccade analysis, AIDEAL treated the conjugate signal, e.g., the L + R eye position/2. The onset and the offset of the saccade were defined as the time points where the peak velocity went above or below 10% of the peak velocity; practically, this corresponded to values above or below 40°/s (as the peak velocity of 20° saccades is typically above 40°/s). The total average velocity was defined as the ratio of total amplitude in degrees divided by time in seconds.

To evaluate binocular coordination of saccades, or the disconjugacy during saccadic movements, the difference in amplitude between the left and the right eye signal was calculated. The disconjugate drift, or the difference in drift amplitude during the first 80 or 160 ms of fixation, was calculated.

### 2.3. Ocular Motor Tests

#### Painting Tests

The data concern eye movements while viewing modern art painting selected because of their specific visual psychophysical attributes: three Op Art paintings from Bridget Riley, providing a strong illusion of visual motion-in-depth, and on the other hand, four paintings from Escher or Magritt with the high complexity of space construction, e.g., bistable spaces or image text contradictions. The reason for choosing such paintings was the hypothesis that dyslexics known to show more unstable fixations in-depth (diverging or converging post-saccadic drifts) would be perceptually more sensitive to the illusion of movement in depth as fixation instability could contribute to the genesis of such an illusion. On the other hand, dyslexics would have a higher capacity in analyzing complex space compositions, or contradictory images. Dyslexic and non-dyslexic teenagers viewed images of each painting for 30 s, each in a standing position in front of a screen, while their eye movements were recorded (with PupiLabs, core device).

Adolescents were asked to stand in front of a laptop that was positioned 40 cm away from their eyes. The laptop was positioned so the image would appear in the middle of their vision. Each adolescent was instructed to try to keep their head still and not to move during the testing. Each participant was asked to fixate at a target on the computer screen at the bottom right corner of the screen before being shown each painting. They then were shown each painting on a black background sequentially for 30 s. Eye movements were recorded during each 30 s session. In between each painting they were given 30 s to rest and reset for the next viewing session. They viewed seven paintings in a row: Bridget Riley, Blaze 1, 1962; Bridget Riley, Movement in Squares, 1961; Bridget Riley, Cataract 3, 1967. The paintings and their size are shown in Table 1

## 3. Pre-Processing and Feature Extraction

### 3.1. Eye Movement Descriptors Based on the AIDEAL Velocity Threshold Method

For each individual and each test, and each of the above-cited features, the AIDEAL software provides the mean value (based on 10 to 20 trials), the standard deviation, the coefficient of the variation (standard deviation/mean × 100), and the total number of trials used for each descriptor. The eye movement features computed are:

Amplitude: the amplitude of the movement (in degrees), i.e., the change in eye position between the onset and the offset of the eye movement; the onset and offset of eye movement being determined by velocity threshold criteria as described above using AIDEAL software.Duration of the phasic component: the time of execution of the movement (in ms), i.e., the time between the onset of the movement and its offset.P-Velocity: Peak velocity of the eye movement (in degrees per sec); the peak velocity is usually achieved at the first 3rd of the trajectory of the movement.A-velocity: Average velocity, which is obtained via the ratio of the amplitude of the movement (in degrees) over its duration (in seconds).Drift 1: is the amplitude (in degrees) of disconjugate drift of the eyes (i.e., subtraction amplitude of the drift of the right eye from that of the left eye) for the 80 ms period starting from the offset of the movement.Drift 2: Is the amplitude of the disconjugate drift during the 160 ms period following the offset of the movement (in degrees).Total Amplitude: The addition of the phasic amplitude and the vergence drift during the 160 msec.Fixation duration: the period (in ms) between two successive fixations.

We considered that outlier values from AIDEAL do not occur randomly and thus convey information on the recorded gaze. For example, aberrant values could be due to blinks or artifacts of eye movement recording apparatus. We, therefore, chose to give these aberrances a value of 0 instead of the population average. Finally, we applied the procedure described in the next section to compute the disconjugacy index.

### 3.2. Disconjugacy Global Index-New Analysis (DGI)

Figure 1 illustrates the nature of the selected segment by our algorithm. The blue curve is the right eye movement over time, while the orange trace shows the movement of the left eye; upward inflexion indicates rightward movement.

The DGI score is a measure of the frequency of the occurrence of the disconjugate segments. A disconjugacy segment is defined as a time interval where two eyes are moving in opposite directions (converging or diverging, as is done when changing depth to look at near vs. far objects). These segments are detected based on the velocity and position of eye movement signals of the two eyes continuously, i.e., the entire time series of signals as opposed to the AIDEAL approach that considers only the movement (saccade or vergence) and the periods immediately before and after the movement (a trial).

The segments extracted by our algorithm are visible in the position trace (Figure 1d,e). our algorithm identifies the segment by searching in the signal of the product of the right and left eye velocity signals; the period during which the product is negative, for example, if the instantaneous velocity of the left eye and right eye is +10°/s and −10°/s respectively, then the product is −100 (°/s)^2^ and this signifies a disjunctive period that we select, in contrast, if the instantaneous velocity is +10°/s and +20°/s for the left and right eye, respectively, then the product is 200 (°/s)^2^ and therefore this segment is not considered in our analysis.

To evaluate the DGI, we counted the total number of the disconjugacy segments, and we divided it by the total duration of the recording. To count the total number of the disconjugacy segments present in a signal, we retrieve the position of the left eye on the horizontal axis over time and the position of the right eye on the horizontal axis over time (1). Then, we computed the disconjugate signal, denoted by D defined as the difference of the left horizontal signal minus the right horizontal signal (2). Also, we computed the derivative of the left signal and the right signals and the product of the two derivatives, denoted by Dv (3). Then, we used the Dv signal to retrieve all the segments where the sign of this signal is negative (4). Those segments are the moments where the two eyes were moving in the opposite direction over the X, horizontal axis, in other words, these are the moments where the two eyes were moving oppositely converging or diverging.

For each Dv segment, we split it into sub-segments whenever the D corresponding block changes its sign (5). It is important to note that considering both the disconjugate position signal and the product of the velocities of the two eyes enables us to identify the periods during which the eyes moving in opposite directions; the D position signal alone does not give this information, as there might be instances where the eyes are traveling in the same direction but by different amounts. Consequently, this new analysis is more focused on identifying continuously, all the instantaneous segments during which the eyes are traveling in opposite directions (therefore we prefer to call this index a distinctiveness signal rather than disconjugate). Once each segment is identified as either micro-diverging or an elementary micro-converging, we retain all those segments that last for a period of at least three samples i.e., 12 ms of time. This decision was based on a prior analysis testing different segment minimal periods and seems physiologically plausible and not just a noise in the eye-tracking signals.

Finally, we counted the total number of the disconjugacy segments, and we divided by the total duration of the recording to make this new descriptor comparable across recordings of different durations (6).

The different steps to compute the DGI score are formally presented in the equation below.
(1)$$X_{l}=\left[l_{1},l_{2},\ldots ,l_{n}\right],\;\;X_{r}=\left[r_{1},r_{2},\ldots ,l_{n}\right]\;\;$$
(2)$$D={\left[X_{l}^{\left(i\right)}-X_{r}^{\left(i\right)}\right]}_{i=1}^{n}$$
(3)$$DV={\left[\frac{X_{l}^{\left(i+1\right)}-X_{l}^{\left(i\right)}}{T^{\left(i+1\right)}-T^{\left(i\right)}}\times \frac{X_{r}^{\left(i+1\right)}-X_{r}^{\left(i\right)}}{T^{\left(i+1\right)}-T^{\left(i\right)}}\right]}_{i=1}^{n}$$
(4)$$E=\{\left(s,l\right)\in \left[2,n]\times [s+3,n-1\right]|DV^{\left(s-1\right)}>0\;\&\;DV^{\left(l+1\right)}>0\;\&\;\forall i\in \left[s,l\right],DV^{\left(i\right)}<0\}$$
(5)$$Count=\overset{E}{\underset{\left(s,l\right)}{\sum }}card(\{\left(a,b\right)\in \left[s,n\right]\times \left[a,l\right]|(D^{\left(a-1\right)}\neq D^{\left(a\right)})\;\&\;(D^{\left(b+1\right)}\neq D^{\left(b\right)})\;\&\;\forall i\in \left[a,b\right],\;D^{\left(i+1\right)}=D^{\left(i\right)}\})\;$$
(6)$$DGI=\frac{Count}{T^{\left(n\right)}-T^{\left(i\right)}}$$

In addition to the DGI score, we’ve considered two other features: the mean amplitude and the mean duration over the different retrieved segments. Physiologically, it is important to know whether the amplitude and duration of such segments are different for the two populations; moreover, such additional features might increase the generalization ability.

## 4. Model Fitting

We followed the same procedure presented in [4] to train a support vector machine(SVM) with a radial basis function RBF kernel and a logistic regression model to predict dyslexia based on eye movement descriptors extracted from the seven paintings dataset.

In the first analysis, we considered the seven datasets separately. In this analysis, for each dataset, all its samples were collected when each subject was observing the same painting. Hence, the different DGI scores were supposed to be easily comparable. However, the sample size was low (92 samples for each dataset).

In the second analysis, we grouped the seven paintings datasets into two groups. By aggregating the painting 1, 2, and 3, datasets into group 1, and painting 4, 5, 6, and 7 into group 2. Hence, the sample size of each dataset has increased from 92 to 276 and 368 for the first and second dataset, respectively. On the other hand, the variance of the input space has increased.

The paintings datasets were grouped based on their visual psychophysics similarity of the painting table, hence the three datasets of the Op Art paintings from Bridget Riley were grouped into group 1, whereas the other four paintings from Escher and Magritt were combined into group 2.

Finally, we’ve followed the same logic of analysis 2, by combining the seven datasets into one big dataset that contains 92 × 7 observations, to have a more reliable score of the generalizability of our models. By merging the seven datasets into one big dataset, the test size has increased from 20 samples to 140 samples. In this analysis we investigated the ability of each feature set (Aideal eye movement feature and the DGI features) to predict dyslexia without having the information about which painting each participant was looking at when eye movements were recorded.

During the three analyses, we used cross-validation (five-fold). At each fold, the model is fitted to the train set, and its generalization ability is evaluated on the test set. We report the average test performance on the five folds. At each fold, the train set, and test set were first normalized by subtracting the mean of the training set then dividing by the standard deviation of the training set. Since the size of the dataset did not allow for the creation of a representative validation set, we do not perform any hyperparameter tuning and use the default values of the different models in their scikit-learn implementation.

To evaluate our ML algorithms, we used three metrics: accuracy, sensitivity, and specificity. A more complete version of the procedure is presented in [4].

## 5. Results

### 5.1. Predicting Dyslexia Using AIDEAL Based Features of Saccades and Fixations

Figure 2 presents a summary of the accuracy of the two models (support vector machine and logistic regression) when using AIDEAL based features to predict dyslexia on the ten datasets. In addition, in Appendix E (Table A1, Table A2, Table A3, Table A4 and Table A5), we present the full tables of the different metrics scores used for the two models when trained on the AIDEAL eye movement features on analysis 1 (Table A1 and Table A2), analysis 2 (Table A3 and Table A4), and analysis 3. Finally, in Appendix C (Figure A5, Figure A6, Figure A7, Figure A8, Figure A9, Figure A10 and Figure A11), we present the distribution of each of the three features and for each of the 7 paintings datasets.

In the first analysis, when considering painting separately, for all seven of the painting datasets, AIDEAL-based eye movement features were able to predict dyslexia. Except for the first dataset (painting 1), there was no significant difference between the performance of the linear model and the non-linear model.

The best performances were achieved when trained on the painting 2 dataset, with nonlinear SVM, with an accuracy of 86.25%, a sensitivity of 82.5%, and a specificity of 90%. On the other hand, the worst model in terms of the generalization ability was the logistic regression when trained on the painting 1 dataset.

In the second and the third analysis, where we’ve grouped multiple datasets to augment the sample size of each dataset. The accuracy has improved to 85.88% using the SVM model.

Finally, the best performance was achieved when merging all the seven datasets, without considering the nature of the painting that was observed during the dataset acquisition process. This improvement can also be explained by the sample size that has been multiplied by seven when considering combining all the seven painting datasets.

### 5.2. Predicting Dyslexia Using DGI of the Continuous Eye Movement Signal

#### 5.2.1. Different DGI Distributions between Dyslexic and Non-Dyslexics Populations

Before adding the DGI set of features to our ML experimental setup, we conducted some exploratory data analysis, to gain more insight into the distributions of the three handcrafted features, namely, the DGI, the mean duration, and the mean amplitude and their ability to predict dyslexia. Figure 3 presents the histogram of the retrieved segments from the seven painting datasets combined for the dyslexic population and non-dyslexic population.

The left histogram corresponds to the non-dyslexic population, while the right histogram corresponds to the dyslexic population. The two histograms were computed using 100 bins, showing a statistical distribution with two main modes that correspond to the peak of micro convergence and micro divergence amplitude segments.

Most DGI segments have an amplitude less than 0.005°. For each of the 100 bins, their count for the dyslexic population are much higher than for the non-dyslexic population when considering the bins with an amplitude less than 0.005°, and is approximately equal to the non-dyslexic population when considering the bins with an amplitude bigger than 0.005°.

Also, in Appendix D (Figure A12, Figure A13, Figure A14, Figure A15, Figure A16, Figure A17 and Figure A18), we present the histogram of the retrieved segments when considering the seven paintings dataset separately. By analyzing the painting dataset separately, we found the same pattern.

The Mann–Whitney U test results shown in Table 2 indicate statistically significant differences between dyslexic and non-dyslexic populations for all paintings. In addition, we present in the Appendix B, the DGI (Figure A2), the duration (Figure A3), and the amplitude (Figure A4) of each patient and for each painting test. Dyslexic populations are represented by a yellow dot and non-dyslexic by a blue dot. The two lines indicate the mean values of each of the two populations.

#### 5.2.2. Stability of DGI Values over Paintings

Table 3, present the mean values of the disconjugacy velocity index for the dyslexic and the non-dyslexic population for each of the seven painting tests. The mean of the DGI over the dyslexic and non-dyslexic population is quite stable over the seven paintings dataset. Hence one can try to predict dyslexia without considering the seven paintings separately.

For the seven-painting dataset, the mean of the computed DGI for the non-dyslexic population is centered at 13.9 with a standard deviation of 0.72. For the dyslexic population, the mean of the computed DGI is centered at 27.38 with a standard deviation of 0.80 meaning that the variability of the DGI means for these particular seven painting studied is very low compared to the mean value.

In addition, in Figure 4, we present the variability of the DGI score over the seven paintings dataset for each individual. The dyslexic population is represented using the red color and the non-dyslexic population is represented using the blue color. For the two populations, the variability of the DGI over the seven paintings is quite low.

Hence, for the two populations, the variability of the DGI is low for the different paintings; meaning that one can predict dyslexia using the DGI value without taking into consideration the identifier of the painting itself. In other words, we expect that the predictability won’t depend on a particular painting.

#### 5.2.3. ML Results: Predicting Dyslexia Using DGI

For each of the seven painting datasets, and each of the logistic regression and RBF-SVM models, the feature importance of the duration was similar to the DGI, whereas the feature importance of the amplitude was very low compared to DGI and the duration. In addition, the accuracy score didn’t improve when considering the duration and the amplitude, in addition to the DGI to predict dyslexia. This is due to the high correlation of The DGI with the amplitude and the duration. Further details of the features’ importance experiments, are presented in the Appendix F. We considered only the DGI feature for further analysis (analysis 2 and 3).

Figure 5 presents the accuracy of the two models (support vector machine and logistic regression). For the three analyses (for each of the ten painting datasets), the score on the test set was overall the same. In other words, the generalization ability of each of the linear and nonlinear models has slightly improved when considering augmenting the sample size of the used dataset.

The best accuracy was achieved when training in analysis 3, on the combined seven painting dataset with an accuracy of 89.71%, a sensitivity of 92.34%, and specificity of 86.99%.

Similarly, to the previous section, in Appendix F (Figure A19 and Figure A20, Table A6, Table A7, Table A8, Table A9 and Table A10), we present the full tables of the different used metrics scores of the two models when trained on the DGI eye movement features in analysis 1 (Table A6 and Table A7), analysis 2 (Table A8 and Table A9), and analysis 3.

### 5.3. Further Investigation and Analysis

#### 5.3.1. Sample Size Importance for AIDEAL

To compare the generalization ability of each model when trained on AIDEAL deterministic based eye movement features, during each of the three analyses, for each model and each analysis, we aggregated the accuracy of the test set of each dataset by weighted means. The different scores are presented in Table 4.

One can expect that the accuracy of each model would decrease when predicting dyslexia from the AIDEAL eye movement feature without taking into consideration the image at which each subject was looking, since the eye movement feature depends on the trajectory of the eye, which depends on the observed image. Hence, ignoring the observed will increase the complexity of the classification problem.

However, the generalization ability of both the linear and nonlinear model, has increased when considering merging the data of the different painting datasets, from 75.53% and 72.85% in analysis 1 to 85.88% and 83.53% in analysis 2, and finally to 88.24% and 89.41% in the last analysis, respectively. This can be explained by the sample size since the size of the dataset used in analysis 3 was seven times bigger than the one used in analysis 1.

If such dependence is not shown for DGI it is due to the very low feature space dimension.

#### 5.3.2. Investigating the Complementarity of AIDEAL and DGI Based Features

To investigate the complementarity of the two feature sets in predicting dyslexia on the dataset of analysis 3, we retrained the two models to predict dyslexia using the DGI feature in addition to the AIDEAL eye movement features. In Table 5, we present the scores of the linear and nonlinear models when combining AIDEAL and DGI. When combining AIDEAL eye movement features and DGI features, we achieve an accuracy of 90.0% and 87.5%, a sensitivity of 90.0%, and a specificity of 90.0% and 85.5%.

### 5.4. Summary of the Results

When considering augmenting the dataset size by merging the seven painting dataset (the third analysis), we’ve found that both the deterministic approach with the AIDEAL eye movement features, and DGI based approach, can successfully predict dyslexia using both the linear and nonlinear models. Also, their predictability has increased when considering merging multiple datasets and ignoring the painting index feature.

In addition, when combining the AIDEAL eye movement feature and DGI feature, we achieve a better accuracy using the RBF-SVM.

## 6. Discussion

### Physiologic Significance

To summarize the results using AIDEAL, deterministic based features and ML/SVM, logistic regression confirms the power of eye movement features to categorize dyslexics from non-dyslexics even in free viewing exploration of the paintings, that were selected for their physical properties (illusion of movement, space composition complexity). This observation is novel and powerful, confirming that even in free viewing, ludic conditions dyslexic’s eyes move differently. The power of the results may be also related to these specific paintings used that engage psychophysically the visual-motor system of the observers whether dyslexic or not. Also, it has been hypothesized by some authors [7,10] that micro eye movements and fixation instability particularly in-depth (e.g., disconjugacy or disjunctive micro-movements of the eyes) contribute to perceiving or enhancing such movement illusion in depth. The psychophysical and subjective reports of the participants from the present study are further discussed by the authors [5,6]. Similarly, the paintings selected here with spatial complexity are presumably particularly challenging for the observers and their perceptual bistability could hypothetically reinforce some of the eye movement features studied here, those evaluated by AIDEAL, particularly the disconjugacy as well as the DGI features. That is to say that the predictive power of the eye movements features is most likely an interplay with the strength of the paintings’ stimuli selected here [11].

The Second novel finding is the contribution of the novel analysis concerning the feature DGI that selects only instances during which the eyes are making micro-movements considering all such segments regardless of whether they occur during saccades, during post-saccadic fixations, or during resting periods between saccades. Such microanalysis is even more basic, providing powerful predictability both for each painting and for the painting regrouped by category (movement illusion vs. space composition and complexity).

In conclusion, eye movement peculiarities enable us to successfully categorize dyslexics from non-dyslexics when viewing those specifically chosen paintings. Deterministic eye movement features (as classically used in the neurology of eye movements, amplitude duration velocity, disconjugacy, and post-saccadic fixation drifts) as evaluated by AIDEAL, as well as a more global index such as the DGI, seem to have predictive power by themselves, and when combined their predictability increases further.

## 7. Limitations

The DGI feature applies to the whole time series of the eye movement data but it is more fragmented as it considers only very few portions of the signals, that is, the moments when the eyes are traveling in opposite directions. It is also important to note that these instances mostly occur during resting periods between saccades and therefore do not concern active eye movement programming and execution phases.

Therefore, we can conclude there is complementarity and a need of both types of analysis for more successful classification that is physiologically plausible. The DGI feature associated with the specific paintings used here can be an approximate rough tool to evaluate both qualities of binocular conjugate control of the eyes in free painting exploration, perhaps related to the sensitivity of the observers to movement illusion and /or complexity of space composition of the images

The Aideal based methods provided more neurologically grounded categorization on properties of eye movement control and binocular eye movement stability. It remains to be seen whether the DGI feature is powerful in any eye movement condition, including oculomotor tests such as saccade and vergence tests, or including free exploration of narrative paintings or more banal everyday images.

To investigate how the DGI feature is powerful in the saccade and vergence tests, we applied the same method described in analysis 1, but on the saccade and vergence dataset present in the previous analysis [6]. We found that DGI can predict dyslexia from the saccade dataset with an accuracy of 75%, and from the vergence dataset with an accuracy of 80%. The full metrics scores on the test set, when training the SVM and the Logistic regression are presented on Table 6 and Table 7.

Hence, the very high predictability of the DGI is lost, which confirms that the power of the paintings chosen, is responsible for the relevance of the DGI; we predict that paintings without depth illusion or complexity would provide less predictive power to the DGI. A current study with classic paintings, e.g., Le tricheur by Georges de La Tour seems to confirm such a prediction.

In contrast to this literature that focuses on the detection of binocular or conjugate micromovements here [12,13,14,15], we focus on micro-movements in the opposite direction for the two eyes. Also, in the microsaccade literature, data cited above are mostly coming from a fixation task during which the subject has to fixate a target for several seconds typically 15 s.

Moreover, we do not limit ourselves to a particular waveform such as saccade type, the movement can be rapid or drifting. Also, the analysis is done on traces of the eyes continuously during a free exploration task that is self-triggered. In a way this analysis is less deterministic. Yet, the only hypothesis would be that the visual features of the paintings (perspective, overlapping, motion, contrast etc.) would somehow stimulate continuously disconjugate or micro-vergence eye movements; this is in line with prior studies see [16,17].

### Controversy about Eye Movement Abnormalities in Dyslexia

There is a long-standing controversy about eye movement abnormalities and the causal link with dyslexia. The majority of the studies conducted concern reading tasks; some of them claiming that abnormalities of saccades are the cause of reading difficulty [15] others claiming that reading saccade abnormalities are the consequence of reading difficulty [16]. Debating on this issue is beyond the focus of this study; in our prior studies we investigated physiologic aspects of saccades and vergence eye movements independently from reading, and also eye movements during reading. The ensemble of these studies, using reading and non-reading conditions, covering both saccades and vergence eye movements provides a new insight into the physiological aspects of binocular motor control inefficiency in dyslexia; it is important to advance beyond causal controversies as the neurologic substrate of dyslexia remains unclear. It is a physiologic reality that eye movement abnormalities can affect the quality of vision and quality of vision is of major importance for reading especially for young readers.

For instance, problems of binocular eye movement coordination can make the letters and words look blurry or double intermittently and this is deleterious for attention and cognition as has been previously shown in young students with the use of the Stroop test e.g., [17]. Our prior research [4] using machine learning and showing that eye movement abnormalities in teenagers while reading text or during saccade vergence ocular motor tests, are able to successfully predict reading speed, at a rate similar to that of language tests used by speech therapists to screen dyslexia, and provides additional evidence that eye movements reflect and determine the quality of reading in such populations.

The present study strengthens this view as it shows again that assessing eye movement abnormalities in ludic conditions, such as free viewing of paintings, can classify dyslexics from non-dyslexics successfully. Eye movement abnormality features while exploring these paintings are not harmful for the aesthetic experience, but it could be a problem for reading and other school learning activities. As an example, Jainta and Kapoula [18] showed in the past, that during reading there were more fixations and more regressions, as well as a tendency for larger saccade amplitudes in dyslexic children compared with non-dyslexic children. Also, fixation durations were found to be slightly longer in dyslexics. Their results were in line with previous reports [15,16,19]. Larger saccades during reading, going beyond the recognizable letters in the periphery could be a motor control problem, or an attention escape strategy moving away from puzzling letters or words, while during painting exploration the same behavior could be useful, enabling multiple views of different points of the painting. The fact that eye movement parameters during free viewing paintings carry the same dysfunctional characteristics as during reading sheds new light on the intrinsic nature of these problems, regardless of the task. Importantly it shows the context specificity, as the abnormalities can be harmful for reading but not for the aesthetic experience and interaction with artwork.

## 8. Conclusions

In this paper we developed a new algorithm to compute a new descriptor that we called the disconjugacy global index (DGI), then we investigated the complementarity of such features with AIDEAL based features to predict dyslexia. We evaluated the generalization ability when the models were trained on three different conditions, where we had a different sample size and different variance for each feature.

In the first analysis, each model was trained on the seven paintings dataset separately and on the Alouette reading test. Also, in the second analysis, each model was trained on two groups of painting. Finally, in the last analysis, each model has been trained on the entire seven paintings datasets.

For each for the three analyses, we used AIDEAL based features and the DGI feature computed from eye movements to predict dyslexia, in the third analysis we also considered the complementarity between the two sets of descriptors, and we found that we achieved the best score in term of accuracy, sensitivity, and specificity. We found that when combining the seven paintings dataset, we can predict dyslexia with an accuracy of 89.71%, a sensitivity of 92.34%, and a specificity of 86.99% when using the DGI feature. Similarly, we found that we can predict dyslexia with an accuracy of 89.41%, a sensitivity of 88.89%, and a better specificity of 90% when using the AIDEAL based.

Finally, we found that by using the A eye movement features with DGI, we can predict dyslexia with an accuracy of 91.21%, a sensitivity of 90.0%, and a specificity of 92%.

## 9. Patents

Zoï Kapoula has applied for patents for the technology used to conduct this experiment: REMOBI table (patent US8851669, WO2011073288); AIDEAL analysis software (EP20306166.8, 7 October 2020; EP20306164.3, 7 October 2020—Europe). Patent application pending EP22305903.1.

## Figures and Tables

**Figure 1 brainsci-12-01031-f001:**
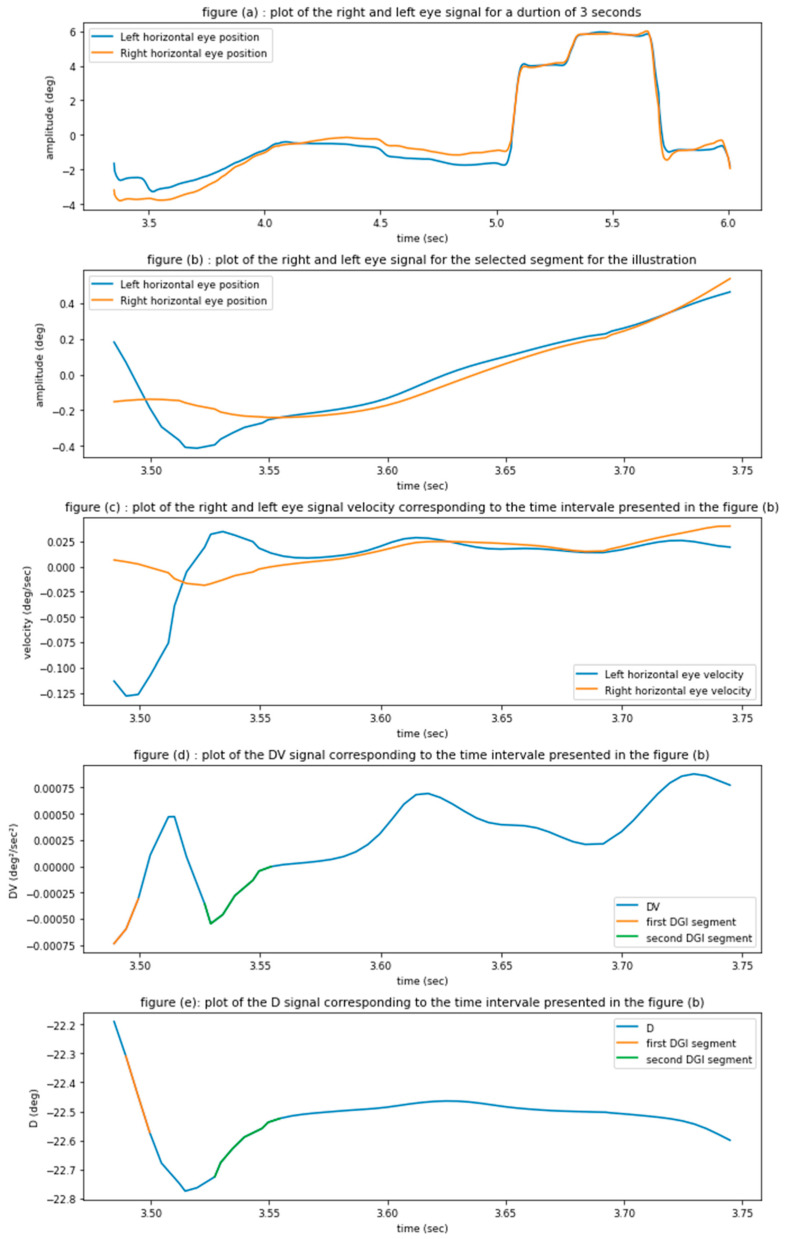
Illustration of the selected segment by our algorithm and the different steps used by the DGI algorithm. Subfigure (**a**) present the time series position of the left eye on the horizontal axis over time and the position of the right eye on the horizontal axis over time. Subfigure (**b**) present a partition of the two signals presented on Subfigure (**a**). Subfigure (**c**) present the first derivative of the two signal presented in subfigure (**b**). Subfigure (**d**) present the DV signal, defined as the product of the first derivatives of the left signal and the right signals. Subfigure (**e**) present the D signal, defined as the difference of the left horizontal signal and the right horizontal signal, and enumerate the DGI segments present on the D signal.

**Figure 2 brainsci-12-01031-f002:**
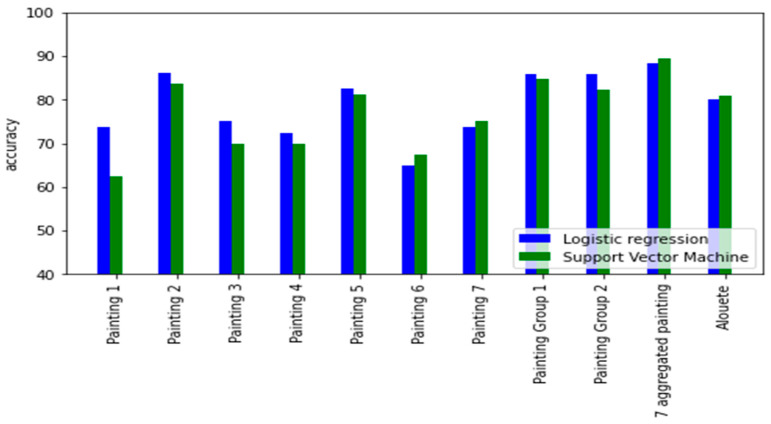
Pairplot of the accuracy of the Support vector machine and the logistic regression when using AIDEAL based features in analysis 1, 2 and 3.

**Figure 3 brainsci-12-01031-f003:**
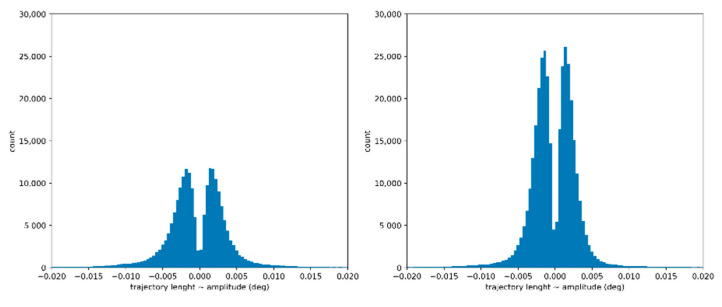
Combined painting—Histogram of the amplitude segments.

**Figure 4 brainsci-12-01031-f004:**
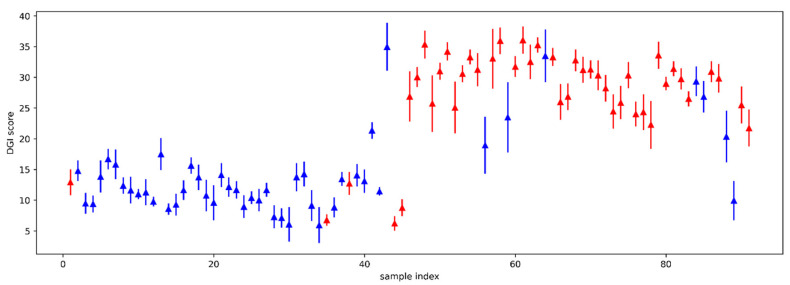
Illustration of the variability of the DGI score over the seven paintings dataset for each patient.

**Figure 5 brainsci-12-01031-f005:**
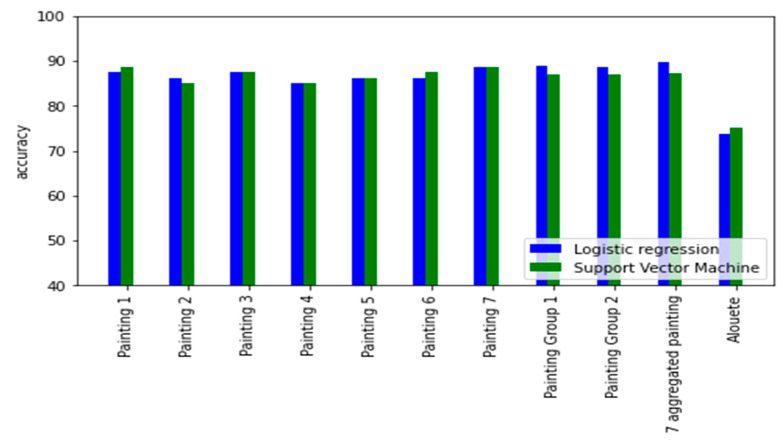
Pair plot of the accuracy of the support vector machine and the logistic regression when using DGI based feature on analysis 1, 2 and 3.

**Table 1 brainsci-12-01031-t001:** Presentation of the seven paintings tables where the children were looking during the data acquisition.

Image	Painter	Title	Size
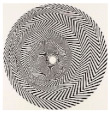	Bridget Riley	Blaze Study (1964)	530 × 521 mm
	Bridget Riley	Movements in square (1961)	122 × 122 cm
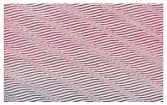	Bridget Riley	Cataract 3 (1931)	221.9 × 222.9 cm
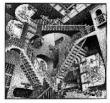	M.C. Escher	Relativité (1953)	27.9 × 28.9 cm
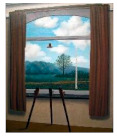	René Magritte	La condition humaine I (1935)	100 × 81 cm
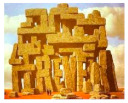	René Magritte	The Art of Conversation-Color Variation (1950)	50 × 65 cm
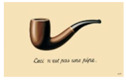	René Magritte	La trahison des images (1973)	60 × 81 cm

**Table 2 brainsci-12-01031-t002:** Mann–Whitney U test for each of the three-parameter and the seven paintings dataset. (*** indicate a *p*-value less than 0.001).

Dataset	DGI	Duration	Amplitude
Painting 1	−3.962 ***	−6.123 ***	−4.034 ***
Painting 2	−4.034 ***	−5.892 ***	−2.811 ***
Painting 3	−2.811 ***	−6.139 ***	−3.907 ***
Painting 4	−3.907 ***	−5.622 ***	−4.352 ***
Painting 5	−4.352 ***	−6.035 ***	−4.733 ***
Painting 6	−4.733 ***	−5.734 ***	−4.280 ***
Painting 7	−4.280 ***	−6.067 ***	−3.962 ***

**Table 3 brainsci-12-01031-t003:** DGI mean values for the dyslexic and non-dyslexic populations for each of the seven paintings datasets.

Dataset	Mean DGI over Non-Dyslexic Population	Mean DGI over Dyslexic Population
Painting 1	14.7	0.28
Painting 2	14.29	0.27
Painting 3	14.23	0.27
Painting 4	14.31	0.27
Painting 5	12.63	0.26
Painting 6	14.1	0.27
Painting 7	13.25	0.27

**Table 4 brainsci-12-01031-t004:** Logistic regression and support vector machine global accuracy scores for the three analyses.

Model	Analysis 1	Analysis 2	Analysis 3
Logistic regression	75.53%	85.88%	88.24%
Support Vector Machine	72.85%	83.53%	89.41%

**Table 5 brainsci-12-01031-t005:** Support vector machine and logistic regression scores when trained on the seven paintings dataset using AIDEAL eye movement and DGI features together.

Model	Accuracy	Sensitivity	Specificity
Support vector machine	90.0%	90.0%	90.0%
Logistic regression	87.5%	90.0%	85.5%

**Table 6 brainsci-12-01031-t006:** Support vector machine scores when using the DGI feature on the saccade and vergence dataset.

Dataset	Accuracy	Sensitivity	Specificity
vergence	75.0%	83.33%	63.33%
saccade	80.0%	86.67%	73.33%

**Table 7 brainsci-12-01031-t007:** Logistic regression scores when using DGI feature on the saccade and vergence dataset.

Dataset	Accuracy	Sensitivity	Specificity
vergence	75.0%	80.0%	66.67%
saccade	80.0%	86.67%	73.33%

## Data Availability

The datasets generated during and/or analyzed during the current study are available from the corresponding author on reasonable request.

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
