# Peer review of "Predicting Dyslexia in Adolescents from Eye Movements during Free Painting Viewing"

_brainsci, 2022, doi:10.3390/brainsci12081031_

Round 1

Reviewer 1 Report

The research is well conducted and the analyses are appropriate.

I would like to read in the discussion the hypothetical neurophysiological mechanisms of the discrepancy between dyslexic and normal children

Author Response

Thank you for this suggestion. We added a paragraph at the end of the discussion about neurophysiologic mechanisms and reading difficulties (see comments to reviewer one)

In addition, we’ve removed the reading analysis from the Result section. We’ve also made minor modifications on  the methods section. all those modifications highlighted using the red color in the article. 

Reviewer 2 Report

The objective of the present study is to explore in how far eye movement features during the visual exploration of paintings can predict dyslexia. Methods and results are well documented, but I have some concerns:

Lines 36-37 and 40-41: In the „introduction the authors claim „… that eye movement abnormalities are intrinsic and are not just a consequence of the reading difficulty“ (36-37) … and „that reading text difficulty can exacerbate an otherwise fragile eye movement control system“ (40-41). It has been demonstrated in many studies that abnormal eye movements are common in readers with dyslexia (e. g. Pavlidis, G.T. 1981, 1985, Rayner, K., Pollatsek, A. 1982; Rayner, K. 1985; Eden, G.F.F., Stein, J.F.F., Wood, H.M.M., Wood, F.B.B. 1994; Hyönä, J., Olson, R.K. 1995; Hutzler, F., Kronbichler, M., Jacobs, A.M., Wimmer, H. 2006; Stein, J. 2018; Blythe, H. I., Kirkby, J.A., Liversedge, S. P. 2018; Biscaldi, M., Gezeck, S., Stuhr, V. 1998; Biscaldi, M., Fischer, B., Hartnegg, K. 2000; Fischer, B., Hartnegg, K. 2000; De Luca, M., Di Pace, E., Judica, A.,   Spinelli, D.,  Zoccolotti, 1999; Trauzettel-Klosinski, S., Kloitzsch, A.M., Dürrwächter, U., Sokolov A.N., Reinhard. J. 2010; Seassau, M., Gerard, C.L., Bui-Quoc, E., Bucci, M.P. 2014). Some authors argue that reading disorders are a consequence of a lack of control of eye movements, others regard unusual eye movements as a consequence of an impaired reading ability. The authors of the present paper cite only some of their own studies but ignore the whole buk of literature on eye movements in reading. The authors should discuss their own research in the light of the present state of scientific knowledge.

Abnormal eye movements during reading are not necessarily a sign of a fragile eye movement control system. No causal relationship between unusual eye movements and poor reading capacity has been substanciated. It has been demonstrated that children with dyslexia can learn adequate eye movements for reading in a single session (Werth 2018, 2019). This shows that these children can perform adequate eye movements and that there is no inability to control eye movements that are needed for correct reading.

Lines 45-46: The claim „there is an interaction between the type of reading text and the eye movement abnormality could be understood as claiming a causal relationship that has not been proven.

Lines 47- 62: Since there is a correlation between eye movement abnormalities and poor reading performance, eye movement features can have a predicting capacity for impaired reading performance as the authors suggest (47-62).

Lines120-126: I suggest to describe the results of the reading tests here, and to provide information about the extent (standard deviations, percentile) to which the reading performance of subjects with dyslexia deviated from the performance of typical readers.

Lines133-439: Methods, evaluation of data, statistics and results are described in detail.

Lines 440 ff: For the "discussion", what has been said before applies: all literature on the role of eye movements in reduced reading performance has been ignored. Again, it should be made clear that no causal relationship between the eye movement features found in the current study and dyslexia has been demonstrated.

The question arises if the results of the study offer any advantage over previous reading tests. Eye movement tests require technical effort and a sophisticated analysis of the data. In view of the fact that unusual eye movements were not demonstrated to be the cause of reduced reading performance, but that only a correlation is demonstrated, no advantage over the usual reading tests can be recognized.

Round 2

Reviewer 2 Report

Introduction and Discussion: The question in how far eye movents can influence reading performance, and in how far eye movements in non reading tasks can shed light on reading problems is not made sufficiently clear. Short fixation times may, e. g. occur in reading and non-reading tasks and may indicate small attention span and/or high distractability of attention. Saccades that exceed the number of simultaneously recognizable letters may be a sign of an increased eye movement impulse, insufficient control of eye movements, or may be due to an insufficient adjustmant of saccade amplitudes and the number of simultaneously recognizable letters with normal eye movement control. Unusual eye movements in dyslexics may correlate with eye movements when viewing pictures with or without a causal relationship. Could you please argue how your findings impact on the reading process and could you please take more into account the current literature on this topic. 

Patients and methods: Could you please describe the kind in which dyslexia was tested, and how much the reading performance of dyslexics deviated from the performance of typical readers.
